# Improved Surface Electromyogram-Based Hand–Wrist Force Estimation Using Deep Neural Networks and Cross-Joint Transfer Learning

**DOI:** 10.3390/s24227301

**Published:** 2024-11-15

**Authors:** Haopeng Wang, He Wang, Chenyun Dai, Xinming Huang, Edward A. Clancy

**Affiliations:** 1Department of Electrical and Computer Engineering, Worcester Polytechnic Institute, Worcester, MA 01609, USA; hwang9@wpi.edu (H.W.); xhuang@wpi.edu (X.H.); 2School of Biomedical Engineering, Shanghai Jiao Tong University, Shanghai 200241, China; chenyundai@sjtu.edu.cn

**Keywords:** electromyogram, transfer learning, deep neural networks, force estimation, CNN, LSTM

## Abstract

Deep neural networks (DNNs) and transfer learning (TL) have been used to improve surface electromyogram (sEMG)-based force estimation. However, prior studies focused mostly on applying TL within one joint, which limits dataset size and diversity. Herein, we investigated cross-joint TL between two upper-limb joints with four DNN architectures using sliding windows. We used two feedforward and two recurrent DNN models with feature engineering and feature learning, respectively. We found that the dependencies between sEMG and force are short-term (<400 ms) and that sliding windows are sufficient to capture them, suggesting that more complicated recurrent structures may not be necessary. Also, using DNN architectures reduced the required sliding window length. A model pre-trained on elbow data was fine-tuned on hand–wrist data, improving force estimation accuracy and reducing the required training data amount. A convolutional neural network with a 391 ms sliding window fine-tuned using 20 s of training data had an error of 6.03 ± 0.49% maximum voluntary torque, which is statistically lower than both our multilayer perceptron model with TL and a linear regression model using 40 s of training data. The success of TL between two distinct joints could help enrich the data available for future deep learning-related studies.

## 1. Introduction

The surface electromyogram (sEMG) has been investigated extensively for myoelectric prosthesis control over the past decades [1]. Studies of sEMG-based myoelectric control can be divided into two categories: pattern recognition-based gesture classification [2] and regression-based force/kinematics estimation [3]. Classification-based control can achieve relatively high accuracy but often has limited gestures available. In comparison, regression can achieve independent simultaneous and proportional control, which allows different degrees of freedom (DoFs) to be active simultaneously [4] with different magnitudes [5,6]. Numerous techniques have been investigated to improve performance, from data preprocessing [1,7] to feature extraction [8,9,10], implementing different machine learning models, including dynamic linear regression [11] and support vector regression (SVR) [12,13].

Recently, deep neural network (DNN) methods have been widely used in sEMG-based myoelectric control [14], often exhibiting better performance than traditional machine learning models [6]. Liu et al. [15] used an artificial neural network (ANN) that only consists of two fully connected layers (i.e., multilayer perceptron (MLP)) to predict dynamic forces. Compared to an MLP using hand-crafted (i.e., researcher-selected) features, convolutional neural networks (CNNs) can learn underlying high-level features from the raw data without much human intervention [16]. Ameri et al. implemented CNN models for regression tasks with raw sEMG as the model input, and the CNN-based approaches outperformed SVR-based schemes [5]. Recently, researchers began to incorporate recurrent neural networks (RNNs) into models, especially long short-term memory (LSTM) networks. Bao [17] used CNN-LSTM regression to predict wrist angles, which showed an advantage over other deep learning models, including separate CNN and LSTM models. However, the performance of deep learning models drops drastically when applied to unseen new user data due to sEMG’s inter-subject variability [18]. This variability is caused by various factors, including different muscle mass, activation patterns, and environment change [19]. Another issue is that data collection for such models can be troublesome [20]; only limited trials can be performed by a subject in a restricted time frame, and limited data will often result in inaccurate representations, leading to overfitting [21].

To solve the issues of lack of generalizability and insufficient training data, transfer learning (TL) can be applied. TL can be defined as follows: given a source domain $D_{s}$ and a learning task $T_{s}$, and a target domain $D_{T}$ and a learning task $T_{T}$, where $D_{S}\;\neq \;D_{T}$ and $T_{S}\;\neq \;T_{T}$, knowledge acquired from $D_{s}$ and $T_{s}$ can be used to improve the learning task $T_{T}$ in $D_{T}$ [22]. TL can help improve generalizability by leveraging source domain data to map the knowledge to the target domain, and, at the same time, solve the issue of limited data for training by aggregating multiple data sources—assuming that the source and target domain share some common knowledge [23,24]. In the field of sEMG-based myoelectric control, TL is generally applied in two different manners: data-driven method or model-driven method. The data-driven method mainly focuses on aligning the data distribution and feature space. Khushaba [25] proposed a framework using canonical correlation analysis to project different users’ data to a unified-style space, which allows it to adapt to the changes from different users with minimal effort. The model-driven method assumes that certain model parameters can be shared between source and target domain models. Du et al. [26] used a CNN model with fine-tuning where weights pre-trained from the source domain were set as the starting point for target domain model training to enhance inter-session gesture recognition. Ketyko et al. [27] proposed a two-stage domain adaptation to improve inter-subject gesture recognition. A transformation matrix was fixed, while an RNN model was trained with source domain data, and then the parameters from the RNN model were transferred to the target domain and transformation matrix, which was subsequently tuned.

While the aforementioned studies overwhelmingly focused on the wrist joint, according to [28], among 10,000 upper limb amputations performed in the United States, 5.2% occurred at the wrist and transradial level, and 6.1% were elbow or transhumeral. These data suggest there is a large amount of elbow joint data left unexplored that could contribute to sEMG-based myoelectric control with TL. Additionally, our previous study [29] has shown that the dynamic relationship between sEMG and force in the wrist joint and elbow joint may be very similar; hence, there should be useful information that can be transferred between these two joints.

Additionally, in studies using TL, the key issue for success is forming a valid and representative source domain dataset [14,18,23,30,31,32]. Two different methods have been used: conducting data collection experiments with human subjects and using open-source datasets. Both methods share a common limitation in the existing literature—when you apply the pre-trained source domain model to a new user’s data, the data must be collected from the same experimental protocol and configuration, i.e., the same number of electrodes, electrode positions, sampling rate, etc. However, there is no standard configuration in sEMG data collection. The number of electrodes varies significantly, from 2 to 16 conventional electrodes to high-density arrays, and the sampling rate ranges from 1000 Hz up to 4000 Hz [7,33]. To the authors’ best knowledge, no existing papers have investigated TL’s performance when the source and target domain data were collected from different joints with different numbers of electrodes and sampling rates.

Based on the issues raised above, in this study, we investigated applying TL to improve sEMG to force estimation using four different deep neural network architectures, where the source and target domain data were collected from two different upper-limb joints (wrist and elbow), with a different number of electrodes and sampling rates. The main contributions of this paper are (i) investigating TL between two different upper-limb joints, (ii) evaluating TL performance when source and target domain sEMG signals were collected from a different number of electrodes and sampling rates, and (iii) direct regression performance comparison of four DNN architectures with sliding window for sEMG to force estimation with/without TL. To the authors’ best knowledge, TL between two different joints has not been presented in the literature, and neither has the optimal sequence length required for DNN architectures in sEMG to force estimations.

## 2. Experimental Dataset

The source domain contains data from 65 subjects collected from four previous experiments [11,34,35,36] where subjects performed elbow extension–flexion contractions. The source and target (see next paragraph) data had been de-identified, and re-analysis of the data did not require human studies supervision, per the WPI Institutional Review Board (File 10-100, 17 November 2017). During the experiments, each subject was seated and secured by belts with shoulder abducted 90°, elbow flexed 90°, forearm aligned in a parasagittal plane, and wrist supinated and cuffed to a load cell for torque measurement (see [11], Figure 1 for a photo of the apparatus). Four bipolar EMG electrodes were secured across each of the biceps and triceps, centered on the muscle midline a quarter of the way to the shoulder insertion from the elbow. The skin above the biceps and triceps was cleaned with alcohol, and electrode gel was used in the two recent studies. Each electrode was 1.75 cm apart and had a pair of hemispherical contacts made of 8 mm diameter stainless steel, separated by 10 mm edge-to-edge and aligned along the long axis of the muscle. A ground electrode was fixed alongside the extension and flexion electrodes. Each sEMG signal was amplified (CMRR > 90 dB at 60 Hz) and filtered by an eighth-order highpass filter at 15 Hz and a fourth-order lowpass filter at 1800 Hz. The sEMG signals and load cell data were sampled at 4096 Hz with 16-bit resolution. Both extension and flexion maximum voluntary torque (MVT) of the elbow were measured. Subjects first warmed up, then raised their force level gradually over 2–3 s to reach MVT, which was maintained for 2 s. Then, 5 s, 50% MVT extension, 50% MVT flexion with constant posture, constant force; and rest (0% MVT) trials were recorded and later used for normalizing gains for sEMG signals. Next, three 30 s constant-posture, force-varying dynamic trials were recorded with three minutes rest between trials to prevent fatigue. Subjects tracked a torque target moving on the screen with feedback signals. The target moved as a bandlimited (1 Hz) uniform random process, ranging from 50% MVT extension to 50% MVT flexion.

The target domain contains data from nine subjects recorded from another previous study [37] where subjects performed hand–wrist contractions. Subjects were seated with elbow supported, shoulder flexed 45° forward along the sagittal plane, and wrist relaxed, while the palm was perpendicular to the floor. A 6-degrees-of-freedom (DoF) load cell was tightly cuffed to the hand for wrist torque measurements. A separate single-axis load cell placed between the thumb and the other four fingers was used to measure open–close torque (see [37], Figures 1–3 for figures depicting the apparatus). The forearm skin was cleaned by alcohol wipes, and electrode gel was applied. Sixteen bipolar electrodes (same specifications as the ones used in the source domain experiments) were secured on the forearm with the middle of the electrode located at 5 cm distal to the elbow crease spread out evenly with equal distance between each electrode. A reference electrode was placed on the ventral forearm. Each sEMG signal was connected to a differential amplifier (CMRR > 100 dB over 30–500 Hz). The sEMG signals and load cell data were sampled at 2048 Hz with 16-bit resolution. After a warmup period, MVT was achieved and measured independently for each direction of the four DoFs (wrist: extension–flexion, radial–ulnar, pronation–supination; hand: open–close). Then, 5 s, 50% MVT contractions for each direction were measured and later used for calibration and normalization. Next, four 40 s constant-posture, force-varying, 1-DoF tracking trials were recorded for each DoF separately, with two minutes rest between trials to avoid fatigue. Subjects tracked a target with feedback forces on the screen. For each DoF, the target moved as a 0.75 Hz bandlimited uniform random process ranging ±30% MVT for each DoF, except for open–close trials, where the range was reduced to ±15% MVT because of the cumulative fatigue found in preliminary testing.

## 3. Methods

### 3.1. Data Normalization and Hyperparameter Tuning

All analyses in this research were performed offline in MATLAB (R2023a, MathWorks, Inc., Framingham, MA, USA) using computational resources supported by the Academic & Research Computing group at Worcester Polytechnic Institute. Raw sEMG signals were normalized by the larger root-mean-square (RMS) value from the 50% MVT trial of the two directions for each DoF, and torque signals were normalized to the MVT of the larger of the two directions per DoF separately for each of the elbow and hand–wrist datasets. Normalizing data can eliminate the difference in scales for force level and make it suitable to aggregate data from different subjects for TL.

Optimal hyperparameters are essential for a neural network model’s performance. Hyperparameter tuning for each neural network model was performed in a “leave-four-out” manner using the elbow dataset: one subject was randomly selected from each of the 4 elbow experiments and set aside, and then the remaining 61 subjects were aggregated and used as training data with the hyperparameters selected to minimize the error on the 4 randomly selected subjects. Training-specific and optimization hyperparameters were finalized as an Adam optimizer and an initial learning rate of 0.001 with a learning rate drop period of 10 epochs and drop rate of 0.2; training data were shuffled after each epoch, with a mini-batch size of 256, and the Glorot initializer was used for random weights initialization. Model-specific hyperparameters are listed in the subsections of each specific model.

### 3.2. Deep Neural Network Architectures

We investigated four DNN architectures on the hand–wrist dataset comprising two feedforward networks, MLP and CNN; and two recurrent networks, LSTM and CNN-LSTM concatenated (C-LSTM). The MLP and LSTM models used hand-crafted features for the networks, while CNN and C-LSTM models used raw sEMG signals as input. All model forms were used to estimate each DoF separately.

#### 3.2.1. Feedforward Neural Networks (FNNs)

The MLP model took sEMG amplitude as feature input. To extract the sEMG amplitude feature (i.e., the time-varying sEMG standard deviation), each sEMG channel was highpass-filtered by a 5th-order Butterworth filter with a 15 Hz cutoff frequency to remove motion artifacts, notch-filtered by a 2nd-order IIR filter (bandwidth ≤ 1.5 Hz) at the fundamental frequency and harmonics to attenuate power-line interference, whitened by a first difference filter [38] to reduce variance, and lowpass-filtered by a 9th-order Chebyshev Type I filter with a cutoff frequency of 600 Hz to remove high-frequency noise, and then rectified, lowpass-filtered at 16 Hz (9th-order Chebyshev Type I), and decimated to 40.96 Hz. A sliding window of 391 ms (16 samples; sufficient to capture the system dynamics [11]) with an increment of 25 ms was applied to each sEMG channel to segment the data (shown in Figure 1), and then, 16 channels were concatenated into one single vector and used as the input for the MLP model. Each sample in the vector was treated as an independent feature in the input layer. Three fully connected (FC) layers (additional layers did not improve model performance) were connected between the input and regression output layers. The last two FC layers were followed by an ReLU activation to introduce non-linearity to the model. The numbers of neurons for the FC layers were 128, 128, and 64, respectively. The structure is shown in Figure 2.

A CNN can extract high-level information from raw sEMG data, producing higher generalizability [31,39]. The fundamental building blocks of CNN models are convolution layers and pooling layers. In convolution layers, the convolution operation extracts information from raw sEMG and then forms feature maps to identify the pattern of each data frame. The outputs of the convolution layers are connected with a non-linear activation function (usually ReLU, due to its simplicity), and then a pooling layer often follows to merge similar features and reduce the dimensionality of the data. The pooling operation is a downsampling process. Raw sEMG data were segmented by a 391 ms (800 samples) sliding window with an increment of 25 ms and used as input for the CNN model. The input for the CNN model was an intuitive sEMG image (800 × 16) representation, i.e., each electrode and time sample was accounted as a pixel of the sEMG image. The CNN model consisted of 4 convolution blocks, each block having a 2D convolution layer with a filter size of 3 by 3, an ReLU activation, and an average pooling layer with a pool size of 2 by 2 and a stride of 2. A batch normalization layer was applied in the first 3 convolution blocks to reduce covariate shift and speed up convergence. Adding batch normalization to the fourth convolution block did not improve the model performance; hence, it was not included due to model pruning. The number of filters for the 4 convolution layers were 64, 128, 128, and 64, respectively. Then, a dropout layer with a dropout rate of 0.5 was applied to avoid overfitting, followed by an FC layer (128 neurons) with ReLu activation connected to the regression output layer (see Figure 3).

#### 3.2.2. Recurrent Neural Networks (RNNs)

Since the sEMG signal is time-series data in its nature [30], many studies claim that a recurrent neural network can help explore the temporal dependencies between sEMG and force, especially with the LSTM network resolving the vanishing gradient problem [40].

An LSTM model was studied. sEMG amplitudes were extracted for each channel as described in the MLP section above. Instead of concatenating 16 channels into one single-feature vector input, for the LSTM model, the input data were treated as a 16-channel sequence input. Each sequence contained data from a sliding window length of 391 ms (16 samples), consistent with the previous models. As shown in Figure 4, the sequence input layer was connected to an FC layer (128 neurons), and then an LSTM layer with 100 hidden units, followed by an FC layer (64 neurons) with ReLU activation, and then connected to a regression output layer.

A C-LSTM model was also evaluated. The CNN model has a limitation in that it cannot explore multiple time windows and the long-term correlation between the sEMG signal and force. LSTM layers can explore and transfer information in different time windows, which can potentially improve force estimation performance [41]. The segmented raw sEMG signals described in the CNN section were used as sequence image inputs for the C-LSTM model. The C-LSTM model (shown in Figure 3) consists of 4 convolution blocks described in the section above with a concatenated LSTM layer (500 hidden units) to express long-term dependence between sEMG and force. Then, a dropout layer (dropout rate of 0.5), FC layer (128 neurons) with ReLU activation, and regression output layer were concatenated.

### 3.3. Performance Evaluation

The hand–wrist dataset contains 9 subjects with 4 different DoFs, each DoF having 4 trials. For each subject and each DoF, we used two trials for training and two for testing, with two-fold cross-validation. Root-mean-square %MVT error (RMSE) for each trial was measured, with the first and last seconds of data omitted due to filter start-up transients caused by the filters used for sEMG amplitude extraction. Even though the CNN and C-LSTM models used raw sEMG signal input and did not have start-up transients, we still omitted the first and last seconds of data for consistency. The average of 4 trials and 4 DoFs %MVT error was used to represent each subject’s performance.

#### 3.3.1. Training Without Transfer Learning (Non-TL)

To evaluate the effect of DNN architectures, we first trained each DNN model without TL using two trials of hand–wrist data and the hyperparameters listed, with two-fold cross-validation. Weights for each layer of the DNN models were randomly initialized, and the models were trained using only target domain (hand–wrist) data. No pre-learned knowledge was available to leverage. Then, we compared the four DNN models’ performances to a conventional linear regression model to determine whether DNN architectures can improve sEMG to force estimation accuracy. Our conventional linear regression model was a 15th-order (equivalent of 391 ms window) FIR model (per sEMG channel), and the parameters were computed via regularized least squares with Moore–Penrose pseudo-inverse, with a singular value tolerance of 0.005 [29].

The initial sliding window length of 391 ms was selected based on previous studies [11,29,37] that used conventional linear regression models. The DNN models may have different characteristics and capture the dynamics differently when compared to the conventional model. Hence, for the linear regression and all DNN models, we investigated eight different window lengths: 98, 195, 293, 391, 488, 586, 684, and 781 ms. The corresponding number of samples per window length varied with the sampling rate.

#### 3.3.2. Transfer Learning (TL)

We aggregated 65 subjects’ data from the source domain (elbow dataset) to pre-train a model for each DNN architecture using the hyperparameters listed above. The first FC layer for the MLP and LSTM model was excluded in the pre-trained model. Then, the weights of the pre-trained models were used as the starting point of the target domain (hand–wrist) DNN models instead of random weights initialization and subsequently fine-tuned with a smaller learning rate (0.0005) using 2 trials of hand–wrist data. Two-fold cross-validation was applied. Due to the high computational cost, only a fixed window length of 391 ms was tested, with this duration based on the results from training without TL (see Results).

For the MLP and LSTM models, the feature input layer and sequence input layer are sensitive to the input size. Both our source domain and target domain sEMG amplitudes were downsampled to 40.96 Hz, but the source domain has 8 channels whilst the target domain has 16. To bridge the dimension difference, a linear transformation matrix was connected to the target domain input layers. The matrix multiplication can be represented as an FC layer; thus, the first FC layer in the MLP and LSTM model served as the transformation matrix to reshape the input size in the target domain, and weights for that layer were randomly initialized and trained from scratch during the TL process since the layer was not included in the pre-trained model.

For the CNN and C-LSTM models, the source domain and target domain sEMG signals have different sampling rates (4096 Hz vs. 2048 Hz) and number of channels (8 vs. 16). Hence, with a sliding window of 391 ms, the sEMG image in the source domain has a dimension of 1600×8, while the sEMG image in the target domain has a dimension of 800×16. However, despite the input dimension difference, we did not need to use a transformation matrix since the convolution kernels are fixed-sized (3 by 3), and they operate through the entire sEMG image in the same way regardless of dimensions.

Then, to demonstrate that TL could reduce computational cost and require less training data, we progressively reduced the training data size for both models that are trained with and without TL. Training/fine-tuning data sizes per trial, separately evaluated, were 40, 35, 30, 25, 20, 15, 10, and 5 s.

### 3.4. Statistics

All statistical analyses used IBM SPSS statistics 28 (IBM, Armonk, NY, USA). Initially, the Shapiro–Wilk test was used to assess the normality of the test results. The Shapiro–Wilk test suggested that all our results followed a normal distribution, so parametric testing was used. A two-way or three-way repeated measures analysis of variance (RANOVA) was used to assess the differences between the performances of the groups of models. When significant interactions were found between the factors, a one-way RANOVA with post hoc pairwise *t*-test with Bonferroni correction (significance level *p* = 0.05) was used to evaluate the simple main effects of each factor at all levels.

## 4. Results

### 4.1. Non-TL DNNs with Varied Sliding Window Length

We investigated eight sliding window lengths on five different non-TL models (linear regression, MLP, LSTM, CNN, C-LSTM). Summary results are shown in Figure 5. A two-way RANOVA found a significant interaction between the model type and sliding window length [F (28,224) = 34.9, *p* < 0.001]. Thus, separate one-way RANOVAs were performed for each model type (factor: window length). Each was significant (*p* < 0.006). The minimum average RMS error (and its corresponding window length) for each model was 6.44 ± 1.07%MVT (684 ms) for MLP, 6.10 ± 1.07%MVT (488 ms) for LSTM, 5.24 ± 0.78%MVT (586 ms) for CNN, 5.09 ± 0.74 (391 ms) for C-LSTM, and 7.15 ± 1.12%MVT (293 ms) for linear regression. The average RMSE for each model trended downward from the smallest window length and then reached a minimum average error at different window lengths, retaining low error thereafter. Post hoc pairwise *t*-tests found that after reaching 391 ms, there were no statistically significant improvements with longer sliding window lengths (*p* > 0.061). Hence, subsequent comparison focused on this window duration.

Figure 6 shows the %MVT error from each model for each individual subject with a 391 ms sliding window. A one-way RANOVA test found statistical differences among the five different model types [F (4, 32) = 74.4, *p* < 0.001]. Post hoc *t*-tests found that the C-LSTM and CNN models did not differ (*p* = 0.438); otherwise, model performance differed statistically in the rank order (best to worse) as follows: C-LSTM/CNN, LSTM, MLP, and linear regression (*p* < 0.008). Similar statistical results among the model types were found within other sliding window lengths, except for 98 ms. An example of true torque and estimated torques from the four DNN models is shown in Figure 7.

### 4.2. Reduced Training Data Size (TL vs. Non-TL)

With the sliding window length fixed at 391 ms, summary results vs. training data size of the two FNN models (MLP and CNN) and, separately, for the two RNN models (LSTM and C-LSTM), each with and without TL, are presented in Figure 8. A three-way RANOVA found significant three-way interactions between model types, with/without TL, and training data sizes [F (21, 168) = 8.4, *p* < 0.001].

Thus, we next pursued a paired comparison between TL and non-TL results. For all four DNN models, at each training data size, training with TL had lower average errors compared to non-TL. Paired *t*-tests (Bonferroni-corrected) found these differences to be statistically significant (*p* < 0.036) for the MLP models with all training data sizes, CNN models with all training data sizes except 40 s, LSTM models with training data sizes from 25 s to 5 s, and C-LSTM models with training data sizes from 15 s to 5 s. Hence, TL was most effective overall at the smaller training data sizes.

Lastly, separate one-way RANOVAs analyzed the effect of training data size for each DNN model, both with/without TL. Training data sizes were significant (*p* < 0.001) for all models, with a trend for error to increase as training data size decreased. Post hoc pairwise *t*-tests (Bonferroni-corrected) compared the full 40 s training data to each other data size. The performance drop became statistically significant (*p* < 0.042) after training data size was reduced to 30 s for MLP with/without TL; 20 s for CNN with TL and 30 s for non-TL; 30 s for LSTM with TL and 25 s for non-TL; and 20 s for the C-LSTM model with TL and 30 s for non-TL. For each model, all subsequently smaller data sizes also had statistically higher errors compared to the error at 40 s (*p* < 0.033)

## 5. Discussion

This study investigated sEMG-based force estimation on a hand–wrist dataset with nine subjects using four different DNN architectures with varying sliding window lengths. We explored the effect of cross-joint TL with reduced training data sizes, where the source and target domain sEMG signals were collected from two distinct upper-limb joints with different numbers of electrodes and sampling rates.

### 5.1. Non-TL DNN Architectures

Initially, we evaluated four DNN architectures and linear regression models without TL with various sliding window lengths (Figure 5). A 391 ms window has previously been shown to be sufficient to capture the dynamics when using a conventional linear regression model [11]. Our study found a similar result for DNN architectures. Model performance improved when initially increasing the sliding window length from 98 ms, then stabilized after 293–391 ms. For all sliding window lengths, all four DNN architectures outperformed the linear regression model.

Next, we compared feature engineering to feature learning with the four DNN architectures (391 ms sliding window). Within the two FNN models, the MLP model used sEMG amplitude as the input (feature engineering), while the CNN model used raw sEMG signals (feature learning). The CNN model had a significantly lower error than the MLP model—a 19% decrease in average %MVT error on a relative scale. Similarly, for the two RNN models, the C-LSTM model, which used raw sEMG signals as input, had a significantly lower error than the LSTM model, which used sEMG amplitude as the input—a 17% decrease in average %MVT error on a relative scale. Hence, feature learning showed a clear advantage over feature engineering. Perhaps hand-crafted features (feature engineering) are highly dependent on pre-processing and human selection, while the CNN blocks (feature learning) can capture high-level features and discover underlying patterns [42]. Another concern is that our feature engineering models only used the sEMG amplitude feature, whereas feature learning models could extract much richer information from the sEMG.

We also compared FNN models vs. RNN models within each feature extraction method. We first looked at the MLP vs. LSTM models since both models used sEMG amplitude as their input. The LSTM model had a 6% lower error (relative scale), which was statistically significant. However, the C-LSTM model did not differ significantly from the CNN model (both feature learning). Thus, the RNN models did not show substantial improvement over the FNN models, possibly because we used overlapping sliding windows to segment the data. The sliding window incorporates data over a short time period and can learn short-term temporal dependencies and less complicated patterns [43], whereas LSTM is more suited for long-term dependencies with complex and varying patterns. These results suggest that there are short-term dependencies (less than 400 ms) between the sEMG signal and force. A sliding window method is sufficient to capture such temporal dependencies, and there did not appear to be any long-term dependencies that require LSTM or other RNN structures, as some studies in the field suggested [44].

Also, our DNN models achieved similar errors compared to the conventional linear regression model but used a shorter sliding window length. For example, the CNN model with a sliding window length of 98 ms had an error of 7.02 ± 0.69%MVT vs. the linear regression model’s error of 7.19 ± 1.13%MVT with a 391 ms window. In real-time systems, a shorter sliding window could lead to smaller buffer sizes, less computation, and fewer time lags.

### 5.2. Transfer Learning

Next, we explored the possibility of cross-joint TL. We aggregated 65 subjects’ data that were collected from the elbow joint, which had a different number of electrodes compared to the hand–wrist data (8 vs. 16) and different sampling rates (4096 Hz vs. 2048 Hz). We used the aggregated data to obtain a pre-trained model for each DNN architecture separately, with the weights from the pre-trained model being used as the starting point and subsequently fine-tuned by the hand–wrist data with a lower learning rate.

In all four DNN architectures, training with TL had statistically lower error compared to training without TL, except at a few selected longer training data sizes. TL required less training data and became more advantageous with less training data. Thus, when training data are limited, TL can possibly improve model performance. The two FNN models experienced more improvement with TL compared to the two RNN models, which again could be explained by the lack of long-term dependencies between sEMG and force.

Compared to cross-subject TL studies where the source domain usually contains 10 to 20 subjects, our cross-joint TL used 65 subjects to form the source domain, which provides more inter-subject variability. The positive transfer results from the elbow joint to the hand–wrist joint suggest that the sEMG-to-force relation is not only similar across subjects (as shown in numerous TL studies) but also across muscle joints. This characteristic provides us with more possibilities for constructing source domain datasets, increasing dataset size, and applying TL.

When compared to the conventional linear regression model, the DNN architectures with TL needed much less training data. With TL, the MLP model with 25 s, the LSTM model with 20 s, the CNN model with 15 s, and the C- LSTM model with 15 s each had lower error than the linear regression model with 40 s of training data.

### 5.3. Limitations and Future Work

There are several limitations in our study. First, when comparing feature engineering vs. feature learning, we only extracted one time-domain feature: sEMG amplitude. Previous feature engineering-based studies [11,45,46,47] have shown that incorporating frequency-domain features, additional time-domain features, and/or (conventional) non-linear models can improve the performance of sEMG-based force estimation using linear regression. Our feature engineering-based models (linear regression, MLP, LSTM) would likely benefit from such methods. However, this comparison is beyond the scope of this study since our main goal was to demonstrate the feasibility of cross-joint TL. In some applications, it may be more desirable to hand-craft all features to permit faster model training (feature engineering), whereas other applications might benefit from spending no time to create features but accept longer model training times (feature learning). Second, our source and target domain sEMG signals were collected when subjects were performing efforts at a similar bandlimited frequency (around 1 Hz). It is unknown whether cross-joint TL is applicable when the source and target domain efforts occur at very different frequencies. Third, both our source and target domain sEMG signals were collected from sparse sEMG electrodes (8 and 16). Cross-joint TL using a high-density sEMG array could be more complicated. Fourth, our study of sliding window length was specific to the EMG-force problem studied. The optimal sliding window length likely varies depending on the specific application, which can range from myoelectric control to ergonomics evaluation to clinical biomechanics to gait analysis, etc.

For future work, we can expand cross-joint TL to two-degrees-of-freedom movement and to limb-absent subjects. We could also convert our regression networks to classification networks, test them on open-source datasets, and compare the performance to other state-of-the-art techniques. Additionally, as more calibration data become available for a given sEMG-force model, more advanced deep learning models can be explored. Such models can lead to lower estimation errors but require more data to avoid overfitting to the larger number of model parameters.

## 6. Conclusions

In this study, we evaluated four DNN architectures with/without TL for sEMG-based force estimation on hand–wrist data. First, we demonstrated that our four DNN models outperform the conventional linear dynamic regression model, and the models using feature learning had better performance than the models using feature engineering. Second, it is shown that a sliding window of 391 ms is enough to capture the sEMG to force temporal dependencies for all DNN architectures tested. Thus, a recurrent network may not be necessary for force estimation; an FNN with a sliding window is sufficient and effective. Third, we proved that TL can be applied when source and target domain sEMG signals were collected from different joints, with different numbers of electrodes and sampling rates suggesting that the sEMG-to-force relationship may be similar across not only subjects but also joints. Lastly, cross-joint TL can improve estimation accuracy and require less training data. In summary, our work provides new possibilities for sEMG-based TL studies; cross-joint TL with different DNN architectures can be expanded to other muscle joints, hand-gesture classification studies, and potentially more.

## Figures and Tables

**Figure 1 sensors-24-07301-f001:**
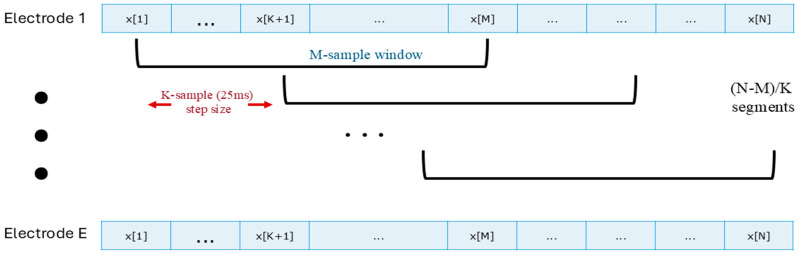
Data segmentation with a sliding window. Symbol $x\left[\cdot \right]$ represents the sEMG amplitude, and N is the total number of samples in each sequence. Each electrode was segmented individually with an M-sample window. (For 391 ms window at 40.96 Hz sampling rate, M = 16.) The consecutive windows are overlapping, with an increment step size of 25 ms (1 sample at 40.96 Hz).

**Figure 2 sensors-24-07301-f002:**
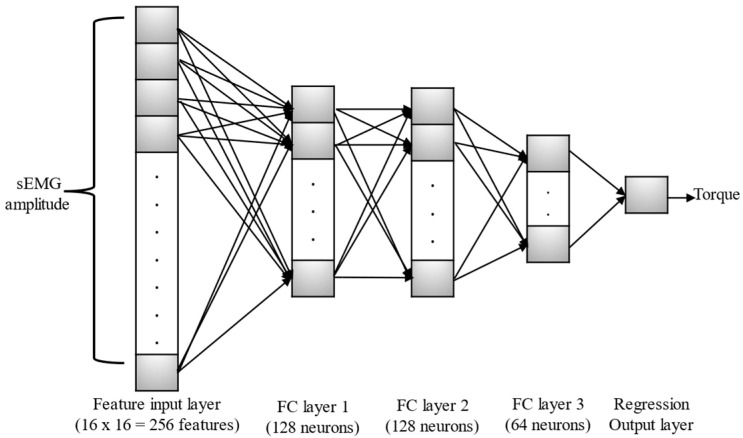
MLP model structure. Feature input layer contains concatenated sEMG amplitudes from 16 channels, each at 16 sample times, then three hidden layers. Each square represents a single neuron. Regression output layer estimates torque.

**Figure 3 sensors-24-07301-f003:**
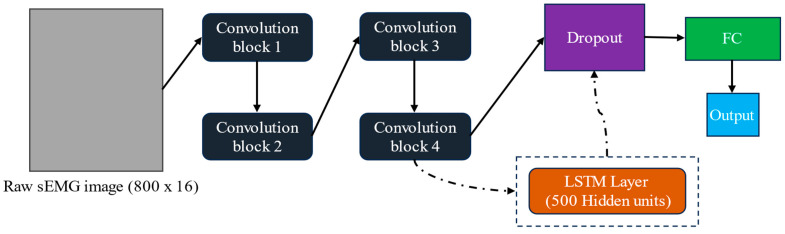
CNN and C-LSTM model structure. Raw sEMG image (800 × 16) was used as the input. Four convolution blocks each comprised *N* convolution filters, batch normalization (except for block 4), ReLU activation, and average pooling. The number of convolution filters in convolution blocks 1 through 4 were 64, 128, 128, and 64, respectively; each filter was of size 3 by 3. There was no LSTM layer in the CNN model; instead, the dropout layer was directly connected to convolution block 4. For the C-LSTM model, the LSTM layer was inserted after convolution block 4, and then the dropout layer followed the LSTM layer.

**Figure 4 sensors-24-07301-f004:**
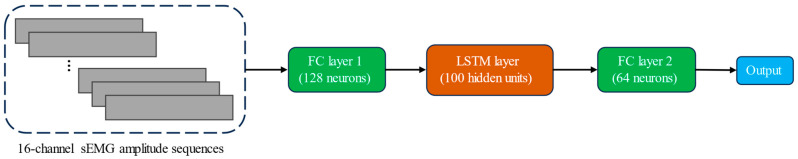
LSTM model structure. Sixteen-channel sEMG amplitude sequences were used as the input. Each sequence had 16 samples. Three hidden layers consisted of an FC layer (128 neurons), an LSTM layer (100 hidden units), and another FC layer (64 neurons).

**Figure 5 sensors-24-07301-f005:**
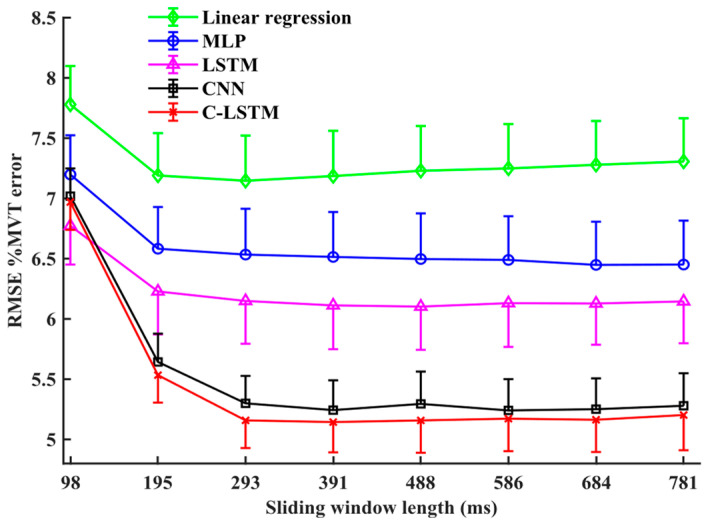
Average %MVT error of 9 subjects for each model vs. the 8 sliding window lengths. Model types are listed in the legend. Single-sided standard error bars are added to each model. Solid lines are an aid for the eye only.

**Figure 6 sensors-24-07301-f006:**
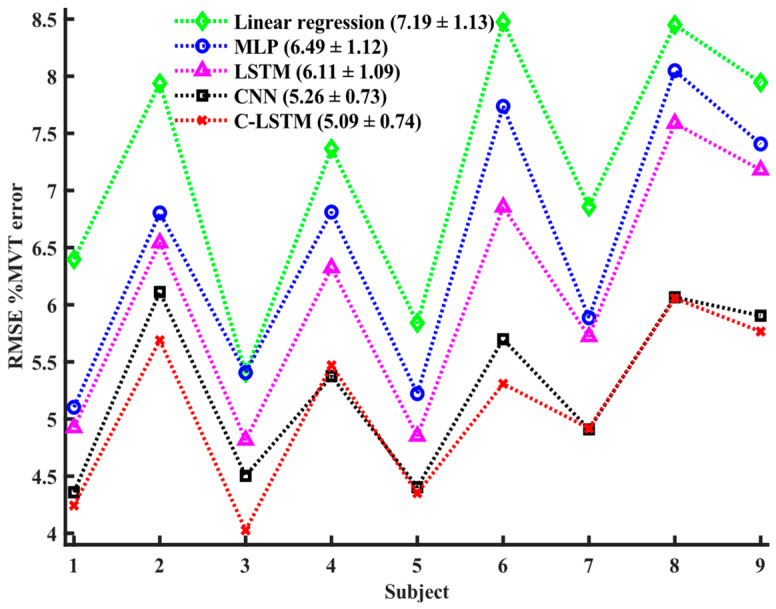
%MVT error for each subject for 4 DNN architectures and linear regression model with a 391 ms sliding window. Model types with corresponding mean ± std %MVT errors are listed in the legend. The dotted lines in between data points are only provided for better visual effects; there are no actual data points on the line.

**Figure 7 sensors-24-07301-f007:**
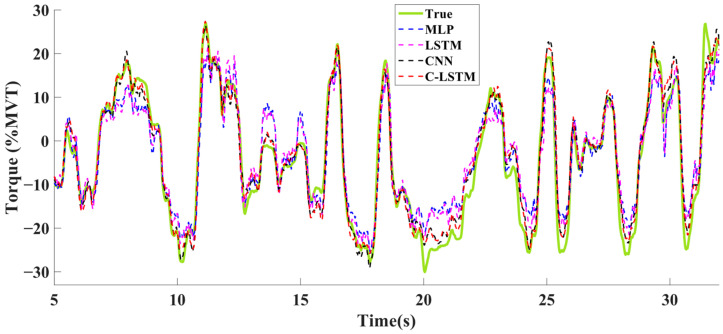
An example segment of true torque and estimated torque from a hand–wrist extension–flexion trial. The solid green line represents the true torque. The blue, magenta, black, and red dash lines represent estimated torques from the MLP, LSTM, CNN, and C-LSTM models, respectively.

**Figure 8 sensors-24-07301-f008:**
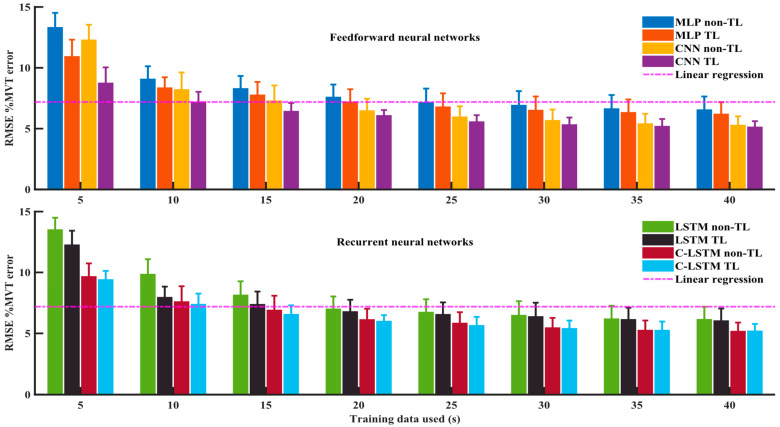
Average %MVT error from 9 subjects for DNN models, with and without transfer learning, with 8 different training/fine-tuning data sizes, from 40 s to 5 s. The top plot contains two FNN models, and the bottom plot contains two RNN models. Model types are listed in the legends. Standard deviations are added to each training data size and each model. A horizontal magenta line represents linear regression model with full (40 s) training data.

## Data Availability

Data are contained within the article.

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
