# Peer review of "Improved Surface Electromyogram-Based Hand–Wrist Force Estimation Using Deep Neural Networks and Cross-Joint Transfer Learning"

_sensors, 2024, doi:10.3390/s24227301_

Round 1

Reviewer 1 Report

Comments and Suggestions for Authors

The authors investigated if pretrained DNN model from elbow can be applied for cross joint force estimation using transfer learning. The research is well conducted and the manuscript is well structured. 

A couple questions may be interested by readers would maybe improve the manuscript

1. in the paragraph from line 344. the authors have conducted comparison between TL and non-TL results, and concluded that TL have overall improvement over non-TL. 

But for CNN with 40s training data and C-LSTM with 30s training data, then it is suggested that non-transfer learning (train from scratch) actually don't differ much. 

I think it should also be noted with some P values.

2. The authors have cited previous work suggesting that 391ms window is enough to capture system dynamics and showed the results in figure 5 as well.

I just have a question for practical application, what is the sliding window best for real-time applications?

Author Response

Reviewer #1:

The authors investigated if pretrained DNN model from elbow can be applied for cross joint force estimation using transfer learning. The research is well conducted and the manuscript is well structured.

A couple questions may be interested by readers would maybe improve the manuscript

Comments 1: in the paragraph from line 344. the authors have conducted comparison between TL and non-TL results, and concluded that TL have overall improvement over non-TL.

But for CNN with 40s training data and C-LSTM with 30s training data, then it is suggested that non-transfer learning (train from scratch) actually don't differ much.

I think it should also be noted with some P values.

Response 1: Transfer learning becomes more advantageous when the training data size is limited. Thus, with a larger (sufficient) training data size, we might not see a statistically significant improvement with TL.

We’ve found two missing p-values in the result section, we’ve added them to line 327 and 360.

Comments 2: The authors have cited previous work suggesting that 391ms window is enough to capture system dynamics and showed the results in figure 5 as well.

I just have a question for practical application, what is the sliding window best for real-time applications?

Response 2: The optimal sliding window length can vary depending on the specific application. This paper investigates the sEMG-to-force relationship in general, even though myoelectric control was cited as the most common application, it can also be used in ergonomics, clinical biomechanics, gait analysis, etc.  Thus, other window lengths may be optimal for these other applications.  We have added this limitation to the “Limitations and Future Work” section of our Discussion.

Reviewer 2 Report

Comments and Suggestions for Authors

This study evaluated two feed forward and two recurrent DNN architectures with/without transfer learning, and with feature engineering and feature learning, using sliding windows for sEMG-based force estimation on hand-wrist data. And it arrives some conclusions, such as a recurrent network may not be necessary for force estimation; a FNN with a sliding window is sufficient and effective, and cross-joint TL can improve estimation accuracy and require less training data, etc. From the topic of this research, it should be in the field of computer science, so I think the authors should focus on developing a deep learning based method for sEMG-based force estimation, not just do some evaluation or comparison.

Author Response

Reviewer #2:

Comments: This study evaluated two feed forward and two recurrent DNN architectures with/without transfer learning, and with feature engineering and feature learning, using sliding windows for sEMG-based force estimation on hand-wrist data. And it arrives some conclusions, such as a recurrent network may not be necessary for force estimation; a FNN with a sliding window is sufficient and effective, and cross-joint TL can improve estimation accuracy and require less training data, etc. From the topic of this research, it should be in the field of computer science, so I think the authors should focus on developing a deep learning based method for sEMG-based force estimation, not just do some evaluation or comparison.

Response: This study mainly aims to investigate the feasibility of transfer learning in sEMG-based force estimation between two upper-limb joints. While we agree that developing new deep learning models could potentially improve estimation accuracy, such models will be limited by the dataset size and computation cost, which are common issues in deep learning studies in EMG to force estimation. Comments describing these limitations already appear in the second paragraph of our Introduction (along with appropriate literature references).  And we argue in the manuscript that transfer learning can help us mitigate these issues. Also, to the best of our knowledge, the comparison of the four architectures in cross-joint transfer learning is novel to the literature.

Reviewer 3 Report

Comments and Suggestions for Authors

The paper is well written and the result shown in Figure 7 for real and estimated torque are good arguments to accept as it is.

Author Response

Reviewer #3:

Comments: The paper is well written and the result shown in Figure 7 for real and estimated torque are good arguments to accept as it is.

Response: Thank you for reviewing this work!

Round 2

Reviewer 2 Report

Comments and Suggestions for Authors

   This revised version almost has no difference with the original one.

Author Response

Reviewer 2 Round 2 Comment: This revised version almost has no difference with the original one.

Yes, we agree that very few changes were made in this version.  The limited number of changes were because very few changes were requested by the reviewers. Overall, we received review comments from the Editorial Board Member and three reviewers.  The Editorial Board Member only requested font changes to Fig. 7, which we produced.  Reviewer 1 requested two minor text changes: the inclusion of two missing p-values (which we added to the manuscript) and that we add text noting that our optimal window length of 391 ms might not be optimal for real-time applications (and we added text to address this topic).  Reviewer 3 did not request any changes.

For Reviewer 2, there was really only one change request, which was rather general and written as: “I think the authors should focus on developing a deep learning based method for sEMG-based force estimation….” 

Our reply to this review comment noted: We believe the question reviewer 2 raised requires a new and independent study compared to our current work. Developing a new deep learning-based method for sEMG-based force estimation is not in the scope of this research, and this is not pure computer science field research. The scope of this research is to investigate the feasibility of transfer learning between two upper-limb joints where the sEMG signal was collected using different numbers of electrodes and sampling rates, this is a novel concept in the field of sEMG-based force estimation.

To evaluate this new transfer learning scenario, we believed the best practice is to use proven deep learning structures (the four models used in this paper) in the sEMG field as our baseline model, since if we were to develop a new deep learning model and investigate a new transfer learning scenario at the same time, it’ll be difficult to evaluate the findings and will require an extremely long article to thoroughly examine every aspect of the issue. Thus, we believe that those two topics should be independent studies and should not interfere with each other in the current stage. Now we’ve established the fact that transfer learning can be applied between two distinct upper-limb joints, we could investigate new deep learning structures in the future, but again it is not in the scope of this research and it’ll be impossible to add a thorough investigation to the current manuscript.

Additionally, even though the four deep learning structures we used in this paper have been investigated in the field, our results provided a new argument that contrary to common belief in the field, a recurrent structure does not have significant advantages over a feedforward structure when using overlapping sliding windows in sEMG-based force estimation, and, feedforward structures are generally less computationally expensive compare to recurrent structures, which is more suitable for real-time applications.

In summary, we believe that including new deep learning models is not in the scope of this research and warrants its independent study. However, based on Reviewer 2’s continuing concern, we have added a short addition to the Limitations section of our Discussion in line 457 to 460.

Reviewer #2 Round 1:

Comments: This study evaluated two feed forward and two recurrent DNN architectures with/without transfer learning, and with feature engineering and feature learning, using sliding windows for sEMG-based force estimation on hand-wrist data. And it arrives some conclusions, such as a recurrent network may not be necessary for force estimation; a FNN with a sliding window is sufficient and effective, and cross-joint TL can improve estimation accuracy and require less training data, etc. From the topic of this research, it should be in the field of computer science, so I think the authors should focus on developing a deep learning based method for sEMG-based force estimation, not just do some evaluation or comparison.

Response: This study mainly aims to investigate the feasibility of transfer learning in sEMG-based force estimation between two upper-limb joints. While we agree that developing new deep learning models could potentially improve estimation accuracy, such models will be limited by the dataset size and computation cost, which are common issues in deep learning studies in EMG to force estimation. Comments describing these limitations already appear in the second paragraph of our Introduction (along with appropriate literature references).  And we argue in the manuscript that transfer learning can help us mitigate these issues. Also, to the best of our knowledge, the comparison of the four architectures in cross-joint transfer learning is novel to the literature.